# Sero-epidemiological survey of *Coxiella burnetii* in livestock and humans in Tana River and Garissa counties in Kenya

**Damaris Mwololo**[1]◉, **Daniel Nthiwa**[2,3]◉*, **Philip Kitala**[4], **Tequiero Abuom**[5], **Martin Wainaina**[3], **Salome Kairu-Wanyoike**[6], **Johanna F. Lindahl**[3,7,8], **Enoch Ontiri**[3], **Salome Bukachi**[9], **Ian Njeru**[10], **Joan Karanja**[10], **Rosemary Sang**[11], **Delia Grace**[3,12], **Bernard Bett**[3]

1 Directorate of Veterinary Services, Ministry of Agriculture, Livestock, Fisheries and Cooperatives, Nairobi, Kenya, 2 Department of Biological Sciences, University of Embu, Embu, Kenya, 3 International Livestock Research Institute, Nairobi, Kenya, 4 Department of Public Health, Pharmacology and Toxicology, Faculty of Veterinary Medicine, University of Nairobi, Nairobi, Kenya, 5 Department of Clinical Medicine, Faculty of Veterinary Medicine, University of Nairobi, Nairobi, Kenya, 6 Department of Veterinary Services, Ministry of Agriculture, Livestock and Fisheries, Nairobi, Kenya, 7 Department of Clinical Sciences, Swedish University of Agricultural Sciences, Uppsala, Sweden, 8 Department of Medical Biochemistry and Microbiology, Uppsala University, Uppsala, Sweden, 9 Institute of Anthropology, University of Nairobi, Nairobi, Kenya, 10 Division of Disease Surveillance and Response, Ministry of Public Health and Sanitation, Kenyatta National Hospital, Nairobi, Kenya, 11 Kenya Medical Research Institute, Nairobi, Kenya, 12 Natural Resources Institute, University of Greenwich, Kent, United Kingdom

◉ These authors contributed equally to this work.
* danielmutiso8@gmail.com

**Data Availability Statement:** All relevant data are within the manuscript and its Supporting Information files.

## Abstract

### Background

*Coxiella burnetii* is a widely distributed pathogen, but data on its epidemiology in livestock, and human populations remain scanty, especially in developing countries such as Kenya. We used the One Health approach to estimate the seroprevalance of *C. burnetii* in cattle, sheep, goats and human populations in Tana River county, and in humans in Garissa county, Kenya. We also identified potential determinants of exposure among these hosts.

### Methods

Data were collected through a cross-sectional study. Serum samples were taken from 2,727 animals (466 cattle, 1,333 goats, and 928 sheep) and 974 humans and screened for Phase I/II IgG antibodies against *C. burnetii* using enzyme-linked immunosorbent assay (ELISA). Data on potential factors associated with animal and human exposure were collected using a structured questionnaire. Multivariable analyses were performed with households as a random effect to adjust for the within-household correlation of *C. burnetii* exposure among animals and humans, respectively.

### Results

The overall apparent seroprevalence estimates of *C. burnetii* in livestock and humans were 12.80% (95% confidence interval [CI]: 11.57–14.11) and 24.44% (95% CI: 21.77–27.26),

**Funding:** The sampling of livestock and humans was implemented under the project: Dynamic Drivers of Disease in Africa: Ecosystems, livestock/wildlife, health and wellbeing (grant no. NEJ001570), funded by the Ecosystem Services for Poverty Alleviation, Programme (ESPA). The ESPA programme was funded by the Department for International Development (DFID), the Economic and Social Research Council (ESRC) and the Natural Environment Research Council (NERC) (received by DG and BB). Additional funding for data analysis and manuscript development was provided through the co-infection project: Co-infection with Rift Valley fever virus, *Brucella* spp. and *Coxiella burnetii* in humans and animals in Kenya: Disease burden and ecological factors, funded by the Defense Threat Reduction Agency, contract number HDTRA 11910031 (received by BB). The funders had no role in study design, data collection and analysis, decision to publish, or preparation of the manuscript.

**Competing interests:** The authors have declared that no competing interests exist.

respectively. In livestock, the seroprevalence differed significantly by species (p < 0.01). The highest seroprevalence estimates were observed in goats (15.22%, 95% CI: 13.34-17.27) and sheep (14.22%, 95% CI: 12.04–16.64) while cattle (3.00%, 95% CI: 1.65–4.99) had the lowest seroprevalence. Herd-level seropositivity of *C. burnetii* in livestock was not positively associated with human exposure. Multivariable results showed that female animals had higher odds of seropositivity for *C. burnetii* than males, while for animal age groups, adult animals had higher odds of seropositivity than calves, kids or lambs. For livestock species, both sheep and goats had significantly higher odds of seropositivity than cattle. In human populations, men had a significantly higher odds of testing positive for *C. burnetii* than women.

## Conclusions

This study provides evidence of livestock and human exposure to *C. burnetii* which could have serious economic implications on livestock production and impact on human health. These results also highlight the need to establish active surveillance in the study area to reduce the disease burden associated with this pathogen.

### Author summary

Q fever caused by *Coxiella burnetii* is a significant zoonotic disease that affects wildlife, domestic animals and humans. This study determined the prevalence of antibodies to *C. burnetii* in livestock (cattle, sheep, and goats) and human populations in arid and semi-arid areas of Kenya between December 2013 and February 2014. We also identified potential factors that were associated with exposure among the above-targeted hosts. Results from this study showed considerable exposure in both livestock and human populations. However, human exposure to this pathogen at the household level was not correlated with herd-level seropositivity. Further studies are needed to elucidate the transmission routes of this pathogen among humans.

## Introduction

Q fever (coxiellosis in animals) is a globally distributed bacterial zoonosis caused by *Coxiella burnetii*, an obligate intracellular pathogen that infects multiple hosts including livestock, wildlife, humans, birds, rodents, reptiles, and arthropods such as ticks [1,2]. This disease is endemic in livestock and human populations in Africa, especially in resource-limited rural areas, where domestic species such as goats, sheep, and cattle are the primary reservoirs of *C. burnetii* infections in humans [3]. Infections in humans in many settings occur mainly through inhalation of contaminated aerosols or dust particles [4,5], but may also occur through direct exposure to contaminated birth products, placenta, faeces, or vaginal mucus during parturition and abortions from infected animals, or during animal slaughter [6–8]. Intake of contaminated milk or dairy products is also another source of infection in humans because these products could contain large quantities of *C. burnetii* [9]. The low infectious dose of 1 to 10 bacteria required to establish infections in humans and animals [10,11], together with the ability of this pathogen to live in the environment for a long time ranging from weeks to months, could contribute significantly to airborne transmission [12,13]. Ticks have also been shown to play a role in the transmission [14].

In humans, infections with *C. burnetii* result in a wide spectrum of clinical manifestations with about 60% of individuals with acute Q fever being asymptomatic [2,15]. Individuals with symptomatic acute Q fever commonly manifest a febrile illness associated with headache, chills, dyspnoea, myalgia, cough, chest pain, arthralgia, atypical pneumonia, hepatitis [16,17], and chronic fatigue syndrome [18]. Acute Q fever is also a significant cause of hospitalization as reported in Tanzania [19], Netherlands [20] and Kenya [21]. This could result in the loss of income and increased disease burden in endemic areas. The main causes of hospitalizations in the above-mentioned countries were pneumonia, anaemia, leukopenia, jaundice, high fever, splenomegaly, shortness of breath, hepatomegaly, fatigue, headache and dizziness. It is also estimated that less than five per cent of individuals with acute Q fever may further develop persistent focalized infections (chronic Q fever) that mainly manifest as endocarditis, especially among individuals with pre-existing heart valve diseases [15], expectant women, and those with immune deficiencies [22]. Neurological and cardiac (acute pericarditis) involvement have also been reported in patients with acute infections [2,23], while other individuals may also manifest myocarditis, a rare but severe and life-threatening condition [24]. This pathogen is also classified as a category B bioterrorism agent, even though case fatalities due to Q fever are usually low [16]. In livestock, coxiellosis is typically asymptomatic or subclinical. However, these infections may cause high abortion rates, stillbirths, infertility, weak calves/kids or lambs, and metritis among pregnant animals, especially cattle [7,25]. Co-infections of Q fever with other zoonotic agents such as *Brucella* spp. and Rift Valley fever virus (RVFV) are also common in animals; this could exacerbate the health outcomes of infected animals since these pathogens cause similar reproductive problems such as abortions [26].

Q fever was first reported in Kenya in the 1950s [27], but information on its epidemiology remains significantly low across the country. This disease is largely neglected by both medical and veterinary personnel as evidenced by a lack of data on national incidence estimates as well as an absence of surveillance and control strategies [27]. Q fever is considered a priority disease in the country due to its high epidemic potential [28]. The disease is commonly misdiagnosed at hospitals as its clinical presentation is similar to other fever-causing illnesses. Brucellosis, leptospirosis, staphylococcal infections, dengue fever, enteric fever, rickettsial infections and malaria are common causes of fever in Africa [29,30] and Latin America [31]. Studies on Q fever have been conducted in Kenya, but the majority of these studies targeted either livestock only [26,32,33], humans only [21,34], or ticks as vectors of *C. burnetii* [35] with very few covering both livestock with humans [36,37]. Over the last decade, One Health studies that incorporate humans and livestock and their social and environmental components are increasingly being used as they allow for better estimation of the risk of infectious disease transmission between livestock and humans, besides providing a comprehensive picture of disease epidemiology [38–40]. This approach has been applied in many countries to inform prevention and control strategies of zoonoses [41].

We therefore assessed the seroprevalence of *C. burnetii* in livestock (cattle, sheep, and goats) and human populations in Tana River county. We sampled only humans in Garissa county due to insecurity. This study further identified putative determinants associated with the seropositivity of *C. burnetii* among these hosts. Findings from this study provides a basis for the establishment of integrated livestock-human surveillance in the area.

## Materials and methods

### Ethics statement

The ethical and animal use clearance for livestock sampling was provided by the International Livestock Research Institute's (ILRI) Institutional Animal Care and Use Committee (IACUC) (reference number: 2014.02). Participants also gave oral consent for livestock sampling. We obtained the ethical approval for human sampling from the African Medical and Research

Foundation (AMREF) Ethics and Scientific Review Committee (reference number: P65/2013). The sampled individuals were adequately informed about the study by the clinicians and their participation was voluntary. These individuals also gave written consents for participation in the study. For children aged 5–12 years, written consents for participation were provided by their parents/guardians while for those aged between 13–17 years, we obtained consents from both the children and the parents/guardians. Adults aged >18 years provided their written consents.

## Study area

This study was carried out in Bura and Hola in Tana River county and Ijara and Sangailu within Garissa county. Tana River county borders Garissa county to the northeast (Fig 1). The above study sites were purposefully selected because of good accessibility. Besides, data on the epidemiology of *C. burnetii* are also very limited among livestock and human populations in these rural sites. These sites have been previously described by other studies [42,43]. In brief, Hola and Bura are both settlement and irrigation schemes established respectively in 1953 and 1978. Tana River County has a binomial rainfall that ranges from 300 mm to 800 mm annually. However, areas close to the coast could receive about 1,200 mm each year [44]. The mean daily temperatures range from 32°C to 37°C. In Ijara and Sangailu, pastoral livestock production for food and income subsistence is the main economic mainstay for the local communities. The

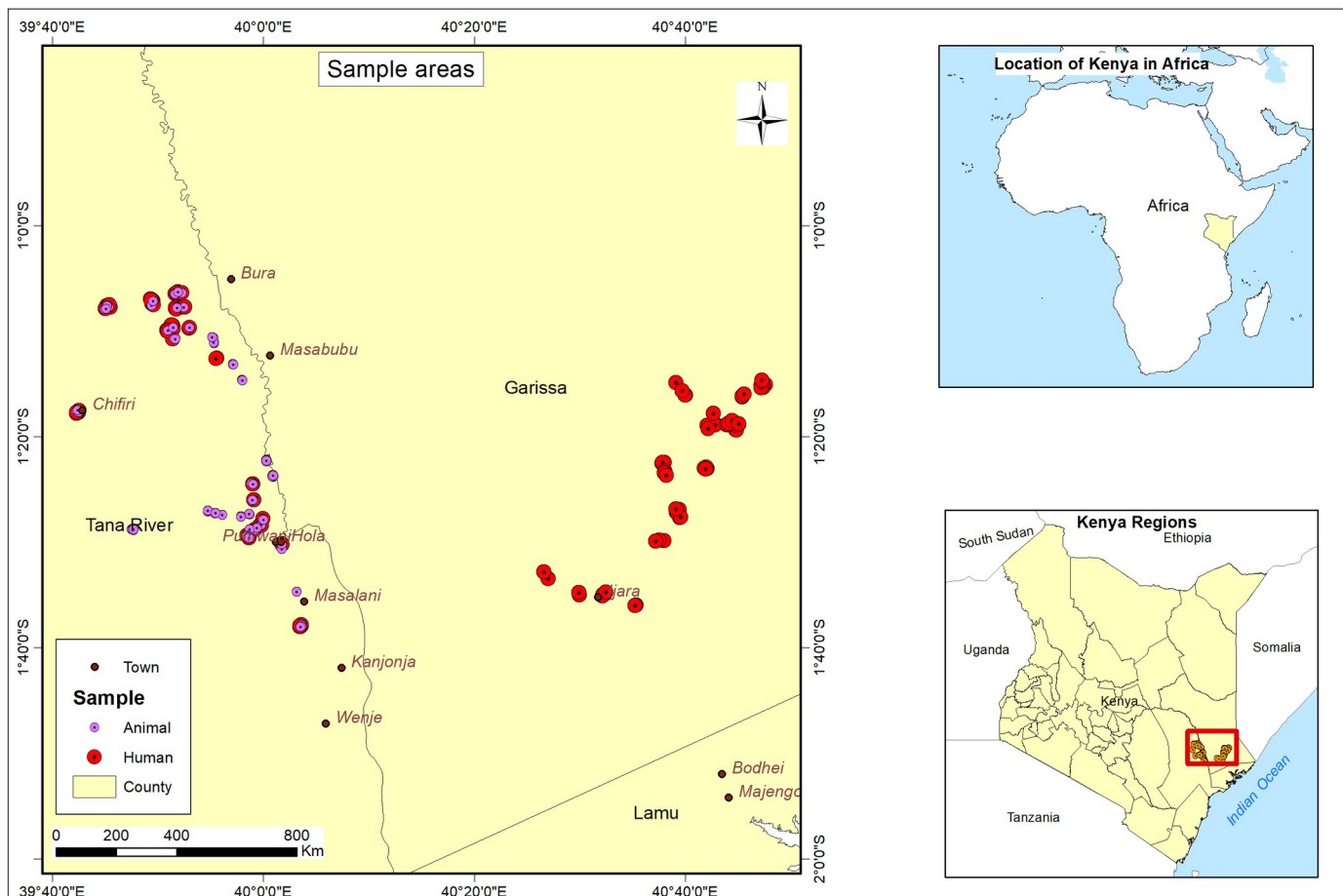

**Fig 1. Map showing sampling sites.** The base layer of the map used to create this figure was downloaded from https://www.diva-gis.org/gdata.

annual rainfall recorded in these sites ranges from 750 mm to 1,000 mm, while the mean daily temperatures range from 34˚C to 38˚C.

## Study design and sample size determination

Livestock and human data were collected between December 2013 and February 2014 through a cross-sectional study with a multistage random sampling design. The sample sizes for livestock (cattle, sheep, and goats combined) and humans in both counties were estimated using the formula; $n = (1.96)^2 p(1-p)/d^2$ [45], with an allowable error (d) of 0.05. The expected seroprevalence estimates (p) of *C. burnetii* in livestock and humans were both taken to be 50% due to limited available data on this estimate in the study area. Given that households were the primary sampling units while animals and humans were secondary sampling units, the estimated sample sizes (n = 384) for livestock and humans were corrected for design effects due to dependence of observations within households (clusters). The design effects (DE) were calculated using the formula: $DE = 1+\rho(m-1)$, where ρ (rho) is the intra-cluster (within-household) correlation coefficient (ICC), and m the number of livestock/humans to be sampled per household [46]. Previous similar studies have used variable ICC values in sample size calculation, for example, 0.1 [40], 0.15 [47] and 0.2 [48], but this study assumed an ICC of 0.3 for both livestock and humans. ICC values typically range between 0.05 and 0.2, although values of more than 0.3 have also been recorded for highly infectious diseases [49]. We chose 20 livestock animals to be sampled per household (cluster) *a priori* due to limited information on herd structure and composition in Tana River County. For the human component of this study, five subjects were to be sampled in each household as we assumed that this was the average number of subjects per household. The estimated design effects for humans and livestock were 2.2 and 6.7, respectively. These design effects yielded an adjusted sample size of 845 for humans, who were to be chosen from 169 households. The required sample size for livestock animals was 2,574; these were to be drawn from 129 households. The required minimum number of households for humans and animals sampling estimated above were derived by dividing their respective adjusted sample sizes with the number of subjects/animals to be sampled per cluster as specified above. Nevertheless, the sampling of animals in each household was proportional to herd size with more animals being sampled in households with more animals. Overall, 2,727 animals that included 466 cattle, 1,333 goats and 928 sheep were sampled from 156 households in Tana River county. A total number of 974 humans (484 in Tana River and 490 in Garissa) were also sampled from 340 households. In Garissa County, the households to be sampled were selected from a list provided by village headmen, while in Tana River, the list was prepared with the assistance of the administrators of Hola and Bura irrigation schemes. In both counties, the households were selected through simple random sampling. Households with at least one of the targeted species (cattle, sheep or goats) were included in the selection criteria since a large percentage (64.6%) of selected households for sampling did not keep all three targeted species. Both livestock and humans were sampled from the same households in Tana River County to allow the estimation of the risk of transmission of *C. burnetii* between these hosts. We did not sample livestock in Garissa County because of insecurity. This is because two successive sampling campaigns were always used, the first covering humans and the second livestock. In this case, the security situation in the area deteriorated soon after we completed human sampling.

## Data collection

A total of 20 randomly selected animals comprising cattle, goats, and sheep were sampled in each selected household. We sampled all animals in households that had less than 20 animals. Blood (10 ml) was collected from the jugular vein of each animal into plain vacutainer tubes and then stored in cool boxes on dry ice. These samples were transported to a local field

laboratory facility for serum separation. The samples were centrifuged at 3000 rpm for 5 minutes and the harvested serum from each animal was stored into 2 ml barcoded cryovials.

Additionally, five humans aged ≥5 years old were also sampled from each selected household. The sampling procedure was conducted by a qualified clinician who was recruited by the project. From each individual, 5 ml blood was taken from the median cubital vein of the left hand into plain vacutainers labelled with unique numbers. The blood samples were taken to the nearby local health centres where they were centrifuged at 3000 rpm for 5 minutes to extract serum. The obtained serum samples were stored in barcoded cryovials. All serum samples were transported on dry ice to the Biosciences eastern and central Africa (BecA) biorepository unit at the International Livestock Research Institute (ILRI) in Kenya, where they were kept at -80˚C until testing for antibodies to *C. burnetii* was performed in 2016. These samples were kept for three years after being collected.

Additional data on the potential factors associated with the transmission of *C. burnetii* in livestock and humans were also collected from each sampled household using structured questionnaires developed using the Open Data Kit (ODK) application. For each sampled animal, we collected data on species (cattle, sheep, or goat), age (calf, kid or lamb, weaner or adult), and sex (male or female). Herd size (the number of cattle, sheep, and goats owned by each household at the time of sampling) was also recorded. The individual-level information collected in the case of humans included gender (male or female), age and occupation. Household-level data such as the source of water, family size and whether the sampled household kept livestock (yes or no) were also captured during sampling.

## Laboratory testing

Serum samples from livestock were tested for phase I and II specific IgG antibodies against *C. burnetii* using a commercial indirect ELISA test kit (IDEXX laboratories, Liebefeld-Bern, Switzerland), following the manufacturers' instructions. The serum samples and controls (negative and positive reference sera) were all tested in duplicates in each test plate and measured the optical densities (ODs) of all wells at 450 nm. The mean ODs of the tested samples and those of positive controls were then corrected by subtracting the mean OD values of the negative controls from each. The percentage positivity (PP) for each tested serum sample was then calculated as; (corrected mean OD 450 of the tested sample/corrected mean OD 450 of the positive control) × 100%. We classified animals with PP of <30% as negative, suspect (borderline) if between 30% and 40% and positive if ≥40% as recommended by the manufacturer.

Human sera were tested for antibodies against *C. burnetii* Phase I antigens using the SER-ION ELISA classic *Coxiella burnetii* Phase I IgA/IgG kit (Virion/Serion, Germany), following the manufacturer's guidelines. The serum samples and reference sera (negative and positive controls) were tested in duplicates for all test plates. The ODs were measured at 450 nm and averaged. The ratio of the OD of the tested serum (S) relative to that of positive control (P) was then calculated using the formula below.

$$Percentage \ \frac{S}{P} = 100 \times \left( \frac{mean \ OD_{450} \ of \ test \ sample - mean \ OD_{450} \ of \ negative \ control}{mean \ OD_{450} \ of \ positive \ sample - mean \ OD_{450} \ of \ negative \ control} \right)$$

We classified humans as positive if the percentage S/P was >10% above the manufacturer's recommended cut-off, suspect (borderline) if +/- 10% of the cut-off, and negative if >10% below the cut-off. The cut-off ranges were calculated by multiplying the average value of the measured standard OD with the numerical data on the quality control certificate. Livestock and human sera with borderline antibody titres were re-tested; samples that gave borderline results after re-testing were considered as negative during data analysis.

## Statistical analyses

All the analyses were performed using R statistical environment, version 3.6.0 [50]. The questionnaire and serological data were imported and merged into one comma-separated value file (.csv). Initial descriptive statistics were generated through cross-tabulations using *gmodels* [51] and *epiR* [52] packages. These analyses included the estimation of the overall apparent seroprevalence of *C. burnetii* with 95% confidence intervals (CI) in livestock and humans. Further estimation of the seroprevalence by each categorical variable was also performed. The categorical variables considered for livestock included species (cattle, sheep and goats), animal sex (male and female), age (calf/kid/lamb, weaner and adult), sampling location (Bura and Hola) and land use type (pastoral and irrigated areas). Herd size (the number of cattle, sheep and goats combined for each household) being a quantitative discrete variable was first tested for the normality of residuals using the Shapiro-Wilk test before it was included in the analysis. This variable violated the assumption of linearity and was therefore converted into a categorical variable with three levels ($\leq$ 20, 21–40 and >40 animals). However, since categorizing a quantitative variable could lead to loss of information as this assumes sudden changes in the response variable at specific values of the independent variable [45], we also compared the results obtained using this categorical variable with those generated using herd size as a log-transformed variable. In addition, due to uneven sex distribution in livestock, the combined data for all animals were also stratified by age to determine the effect of animal sex on the seroprevalence of *C. burnetii*. Seroprevalence estimates were also calculated for each livestock species and by the above independent variables. For human data, we estimated seroprevalence by gender (male and female), occupation (nomadic pastoralists, mixed crop-livestock farmers, students and others [e.g., nurses, housewives, drivers, chiefs, etc.]), land use type (irrigation, pastoral and riverine), source of water (boreholes, canals, dams, taps and others), sampling location (Bura, Hola, Ijara and Sangailu), and ownership of livestock (no and yes). Both age and family size had been recorded as quantitative discrete variables and were also assessed for normality of residuals before being included in the analysis. However, these two variables did not satisfy the linearity assumption. Age and family size were both converted into categorical variable with three levels of $\leq$17, 18–40 and >40 and $\leq$5, 6–10 and >10, respectively. Analysis was also performed using log-transformed formats of these two variables. The choice of the categories for all quantitative discrete variables (herd size, age and family size) was informed by a previous study implemented in the area [43].

The relationships between categorical variables and the seropositivity of *C. burnetii* among livestock and humans were first assessed using the $\chi^2$ test. We also used a subset of human data from Tana River county to determine whether herd-level seropositivity of *C. burnetii* was positively correlated with human exposure. In this analysis, we classified a herd as exposed if at least one animal (cattle, sheep, or goat) within a household had reactive antibodies against *C. burnetii*.

Further analyses of livestock and human data were also performed using univariable mixed-effects logistic regression models to determine the unconditional associations between the outcome variable and the independent variables for each host group. Variables with p values $\leq$ 0.20 by these univariable analyses (for both livestock and human data) were further selected for multivariable analysis [45,53]. Generalized linear mixed-effects models (GLMMs) were used to perform all the univariable and multivariable analyses. We fitted livestock and human data to these models using the *glmer* function in the *lme4* package [54] and adjusted for within-household correlations of observations using the household ID as a random variable. The final multivariable mixed-effects models were selected through a backward stepwise elimination procedure. We first fitted intercept-only null models (without independent

variables), each for livestock and human data, followed by maximal models using respective variables selected from the univariable analyses. The maximal models were then reduced systematically by dropping variables with p > 0.05 until final models with the lowest Akaike's information criteria (AIC) values were selected. Final models were assessed for fit by inspecting plots of residuals versus fitted values obtained from these models [55]. We also tested for potential interaction effects among the independent variables selected in the final multivariable models by creating two product terms for these variables and determined the statistical significance of the main effects using the likelihood ratio test (LRT) [53]. The ICC estimates for within-household clustering of *C. burnetii* exposure among livestock animals and humans were calculated from the variance estimates of the respective final multivariable models using the *icc* function from the *sjstats* package [56].

## Results

### Descriptive statistics

In total, 2,727 animals (466 cattle, 1,333 goats and 928 sheep) sampled in Tana River county were tested for antibodies against *C. burnetii*. The percentage number of animals sampled in Hola and Bura were 79.32% (n = 2163) and 20.68% (n = 564), respectively. Overall, more females (78.14%) were sampled than males (21.86%). The overall median herd size (cattle, sheep and goats combined) was 52 (range: 2–460), while the median herd/flock sizes for cattle, goats and sheep were 35 (range: 2–200), 20 (range: 0–120) and 15 (range: 0–200), respectively. The median number of livestock animals sampled in each household was 11.5 (range: 1–77).

A total of 974 humans (484 in Tana River and 490 in Garissa) were also screened for antibodies against *C. burnetii*. A large proportion of the sampled humans (95.4%) were from livestock-keeping households. The median number of humans sampled in each household was 3 (range: 1–5), while the median age was 26.5 years (range: 5–90).

### Seroprevalence of *Coxiella burnetii*

The overall animal-level seroprevalence of *C. burnetii* in livestock was 12.80% (95% CI: 11.57–14.11). The observed seroprevalence differed significantly by livestock species ($\chi^2$ = 48.80, df = 2, p < 0.01). Goats had the highest seroprevalence of 15.22% (95% CI: 13.34–17.27), and sheep 14.22% (95% CI: 12.04–16.64), and cattle 3.00% (95% CI: 1.65–4.99) followed in decreasing order (Table 1). From the analysis performed using combined data for all animals, female animals (14.50%, 95% CI: 13.03–16.07) had significantly (p < 0.01) higher levels of exposure compared to males (6.71%, 95% CI: 4.84–9.03). However, the results obtained from the analyses done using data stratified by age showed only significant differences in seroprevalence between adult females and adult males ($\chi^2$ = 25.32, df = 1, p < 0.02). We did not observe these differences between female and male weaners or female and male calves/kids/lambs. Adult females and males had seroprevalence estimates of 16.98% (95% CI: 15.23–18.85, n = 1708) and 11.34% (95% CI: 7.9–15.56, n = 291), respectively. Seroprevalence estimates of female and male weaners were 4.62% (95% CI: 2.55–7.63, n = 303) and 2.23% (95% CI: 0.73–5.13, n = 224), respectively. Lastly, the seroprevalence estimates for female and male calves/kids/lambs were 1.41% (95% CI: 0.04–7.60, n = 71) and 2.94% (95% CI: 0.35–10.22, n = 68), respectively. The seroprevalence estimates of *C. burnetii* also differed significantly among animal age groups ($\chi^2$ = 73.45, df = 2, p < 0.01). We recorded higher seroprevalence figures in adult animals (16.16%, 95% CI: 14.57–17.85) than both weaners (3.61%, 95% CI: 2.18–5.57) and calves/kids/lambs (2.16%, 95% CI: 0.46–6.18) (Table 1). The proportion of seropositive animals for *C. burnetii* also differed significantly by sampling locations ($\chi^2$ = 19.47, df = 1, p < 0.01). Hola had a higher seroprevalence

estimate (14.24%, 95% CI: 12.79–15.78) compared to Bura (7.27%, 95% CI: 5.27–9.73). A distribution of the seroprevalence estimates for cattle, sheep and goats with each independent variable as well as results from the $\chi^2$ test are shown in Table 2.

**Table 1. Results of variables assessed for their association with *C. burnetii* seropositivity in livestock using univariable mixed-effects logistic regression models.**

| Variable | Category | n | % Seroprevalence (95% CI) | Odds Ratio (95% CI) | P-value |
|---|---|---|---|---|---|
| Species | Cattle | 466 | 3.00 (1.65–4.99) | 1.00 (Ref.) | |
| | Goats | 1333 | 15.22 (13.34–17.27) | 5.28 (2.80–9.96) | < 0.01 |
| | Sheep | 928 | 14.22 (12.04–16.64) | 5.26 (2.73–10.15) | < 0.01 |
| Sex | Male | 596 | 6.71 (4.84–9.03) | 1.00 (Ref.) | |
| | Female | 2131 | 14.50 (13.03–16.07) | 2.23 (1.54–3.22) | < 0.01 |
| Age* | Calf/kid/lamb | 139 | 2.16 (0.45–6.18) | 1.00 (Ref.) | |
| | Weaner | 527 | 3.61 (2.18–5.57) | 1.47 (0.41–5.25) | 0.55 |
| | Adult | 1999 | 16.16 (14.57–17.85) | 7.29 (2.22–23.99) | 0.01 |
| Location | Bura | 564 | 7.27 (5.27–9.73) | 1.00 (Ref.) | |
| | Hola | 2163 | 14.24 (12.79–15.78) | 2.11 (1.12–4.99) | 0.02 |
| Land use | Pastoral | 551 | 9.61 (7.29–12.39) | 1.00 (Ref.) | |
| | Irrigation | 2176 | 13.60 (12.19–15.11) | 1.77 (0.89–3.49) | 0.10 |
| Herd size** | ≤20 | 620 | 14.35 (11.69–17.36) | 1.00 (Ref.) | |
| | 21–40 | 419 | 12.89 (9.83–16.79) | 0.9 (0.4–1.9) | 0.78 |
| | >40 | 1688 | 12.20 (10.68–13.86) | 0.8 (0.4–1.4) | 0.41 |

n, number of animals sampled in each category; Ref, reference category; CI, confidence interval.

* The sum in age categories was not equal to the total number (n) of animals sampled (2,727) due to missing data (0.02%).

**Herd size comprised cattle, sheep and goats.

**Table 2. Distribution of the seroprevalence of *C. burnetii* by livestock species.**

| Variable | Category | Cattle | | | Sheep | | | Goats | | |
|---|---|---|---|---|---|---|---|---|---|---|
| | | n | % Seroprevalence (95% CI) | P-value | n | % Seroprevalence (95% CI) | P-value | n | % Seroprevalence (95% CI) | P-value |
| Sex | Male | 125 | 0.80 (0.02–4.38) | 0.12 | 206 | 8.25 (4.88–12.88) | 0.01 | 265 | 8.30 (5.28–12.30) | < 0.01 |
| | Female | 341 | 3.81 (2.04–6.43) | | 722 | 15.93 (13.33–18.80) | | 1068 | 16.95 (14.74–19.33) | |
| Age* | Calf/kid/lamb | 77 | 1.30 (0.03–7.02) | 0.00 | 25 | 8.00 (0.98–26.03) | < 0.01 | 37 | 0.00 (0.00–9.49) | < 0.01 |
| | Weaner | 163 | 0.00 (0.00–2.23) | | 155 | 3.87 (1.43–8.23) | | 209 | 6.22 (3.35–10.40) | |
| | Adult | 226 | 5.75 (3.10–9.64) | | 741 | 16.73 (14.11–19.62) | | 1032 | 18.02 (15.72–20.51) | |
| Location | Bura | 245 | 3.67 (1.69–6.86) | 0.61 | 135 | 9.63 (5.23–15.90) | 0.11 | 184 | 10.33 (6.33–15.66) | 0.05 |
| | Hola | 221 | 2.26 (0.74–5.20) | | 793 | 15.01 (12.59–17.68) | | 1149 | 16.01 (13.94–18.26) | |
| Land use | Pastoral | 230 | 2.61 (0.96–5.59) | 0.79 | 114 | 14.91 (8.93–22.80) | 0.78 | 207 | 14.49 (10.00–20.04) | 0.83 |
| | Irrigation | 236 | 3.39 (1.47–6.57) | | 814 | 14.13 (11.81–16.71) | | 1126 | 15.36 (13.31–17.60) | |
| Herd/flock size** | ≤20 | 39 | 2.56 (0.00–13.48) | 0.88 | 141 | 8.51 (4.48–14.39) | 0.10 | 440 | 17.27 (13.86–21.13) | 0.33 |
| | 21–40 | 87 | 2.30 (0.03–8.06) | | 145 | 16.55 (10.90–23.62) | | 187 | 14.97 (10.19–20.91) | |
| | >40 | 340 | 3.24 (1.63–5.71) | | 642 | 14.95 (12.28–17.95) | | 706 | 14.02 (11.55–16.81) | |

n, total number of animals sampled in each category; CI, confidence interval.

* The total number of sheep (921) and goats (1278) under the age categories were not equal to their respective sampled numbers due to missing data.

** The total number of cattle, sheep and goats sampled were 466, 928 and 1333 respectively.

The overall individual-level seroprevalence of *C. burnetii* in humans was 24.44% (95% CI: 21.77–27.26). The seroprevalence estimates in Tana River and Garissa counties were 25.21%

(95% CI: 21.40–29.32) and 23.67% (95% CI: 19.98–27.69), respectively. These estimates did not differ significantly between the two counties (p = 0.58). The results of the independent variables analysed for their associations with *C. burnetii* seropositivity in humans are presented in Table 3. From these analyses, gender was significantly associated with the seropositivity of *C. burnetii* in humans ($\chi^2$ = 4.88, df = 1, p = 0.03). More males (28.27%, 95% CI: 23.81–33.08) were found to be seropositive than females (22.03%, 95% CI: 18.75–25.60). The seroprevalence estimates for Sangailu were the highest at 28.69% (95% CI: 23.02–34.90). Hola (25.68%, 95% CI: 20.06–31.95), Bura (24.81%, 95% CI: 19.70–30.50) and Ijara (18.97%, 95% CI: 14.33–24.36) followed in decreasing order. No statistically significant difference was observed between these sites (p = 0.09). Our results also showed that herd-level seropositivity of *C. burnetii* in Tana River county was not a significant determinant of human exposure (p = 0.19) (Table 3).

**Table 3. Results showing the seroprevalence estimates of *C. burnetii* in humans based on analysis performed using combined data for all animals from both counties and subset data from Tana River county.**

| Variable | Category | Combined data from both counties | | | Tana River county | | |
|---|---|---|---|---|---|---|---|
| | | n | % Seroprevalence (95% CI) | P-value | n | % Seroprevalence (95% CI) | P-value |
| Gender | Male | 382 | 28.27 (23.81–33.08) | 0.03 | 180 | 26.11 (19.86–33.17) | 0.74 |
| | Female | 590 | 22.03 (18.75–25.60) | | 303 | 24.75 (20.00–30.01) | |
| Occupation | Pastoralist | 246 | 26.42 (21.02–32.40) | 0.32 | 145 | 26.90 (19.88–34.89) | 0.53 |
| | Mixed crop-livestock farmer | 170 | 25.29 (18.95–32.52) | | 35 | 17.14 (6.56–33.65) | |
| | Student | 114 | 29.82 (21.62–39.11) | | 29 | 31.03 (15.28–50.83) | |
| | Other* | 73 | 17.81 (9.93–28.53) | | 50 | 22.00 (11.53–35.96) | |
| Age | ≤17 | 339 | 27.14 (22.48–32.21) | 0.30 | 177 | 27.68 (21.24–34.90) | 0.41 |
| | 18–40 | 375 | 22.13 (18.03–26.68) | | 191 | 22.00 (16.33–28.54) | |
| | >40 | 258 | 24.42 (19.31–30.13) | | 115 | 26.96 (19.11–36.03) | |
| Location | Ijara | 253 | 18.97 (14.33–24.36) | 0.09 | | | |
| | Sangailu | 237 | 28.69 (23.02–34.90) | | | | |
| | Bura | 262 | 24.81 (19.70–30.50) | | 262 | 24.81 (19.70–30.50) | 0.83 |
| | Hola | 222 | 25.68 (20.06–31.95) | | 222 | 25.68 (20.06–31.95) | |
| Own livestock | No | 33 | 30.30 (15.59–48.71) | 0.47 | 14 | 35.71 (12.76–64.86) | 0.36 |
| | Yes | 691 | 24.75 (21.57–28.14) | | 455 | 24.84 (20.93–29.07) | |
| Land use | Irrigation | 268 | 26.87 (21.65–32.60) | 0.54 | 8 | 0.00 (0.00–36.94) | 0.25 |
| | Pastoralism | 628 | 23.41 (20.15–26.92) | | 428 | 25.47 (21.40–29.87) | |
| | Riverine | 74 | 24.32 (15.10–35.69) | | 48 | 27.08 (15.28–41.85) | |
| Source of water | Borehole | 127 | 20.47 (13.83–28.54) | 0.36 | 84 | 15.48 (8.51–25.00) | 0.18 |
| | Canal | 231 | 28.14 (22.43–34.41) | | 231 | 28.14 (22.44–34.41) | |
| | Dams | 336 | 23.81 (19.36–28.73) | | 127 | 24.41 (17.23–32.82) | |
| | Tap | 13 | 30.77 (9.09–61.43) | | 12 | 33.33 (9.92–65.11) | |
| | Others e.g., rivers | 17 | 35.29 (14.21–61.57) | | 15 | 33.33 (11.82–61.62) | |
| Family size | ≤5 | 212 | 27.83 (21.91–34.38) | 0.13 | 96 | 19.79 (12.36–29.17) | 0. 13 |
| | 6–10 | 497 | 25.35 (21.58–29.42) | | 285 | 29.12 (23.91–34.77) | |
| | >10 | 263 | 20.15 (15.48–25.52) | | 103 | 19.42 (12.28–28.38) | |
| Herd exposure | Exposed | | | | 181 | 28.33 (21.88–35.52) | 0.19 |
| | unexposed | | | | 138 | 21.58 (15.06–29.35) | |

n, number of humans tested; CI, confidence interval. Except for location, all the other variables in this table (gender, occupation, age, own livestock, land use, source of water and family size) had missing data. Thus, the totals within their respective categories were not equal to the sampled number of 974 for the combined data or 484 for Tana River county. Herd exposure also had missing data.

*Other occupations included community health workers, businessmen and women, housewives, chiefs, drivers and those employed such as nurses.

### Analysis of factors associated with the seropositivity of *C. burnetii* in humans and livestock

**Univariable analyses.** The results of the univariable models (with households as random effects) used to analyse livestock data are shown in Table 1. These results showed that both sheep and goats had significantly higher odds of exposure to *C. burnetii* compared to cattle. With regards to animal sex, we found higher odds of seropositivity for *C. burnetii* among female animals than males. Considering the age variable, only adult animals were found to have higher odds of seropositivity compared to calves/kids/lambs. We observed significantly higher odds of seropositivity of *C. burnetii* among animals raised in Hola compared to those raised in Bura. Both herd size (as a categorical and log-transformed variable) and land use type were not significantly associated with *C. burnetii* seropositivity in animals and were excluded in the multivariable analysis. In the case of human data, the univariable mixed-effects models (results not presented) yielded comparable results to those obtained using the $\chi^2$ test in Table 3. Furthermore, we did not find significant associations between age and family sizes and the seropositivity of *C. burnetii* in humans. Age and family size were both included in the univariable models in turns as categorical and log-transformed variables.

### Multivariable analyses

The results obtained from the final multivariable mixed-effects model used to analyse livestock data are summarized in Table 4. These results showed female animals to have 1.65 higher odds of being seropositive than males. In addition, both sheep and goats had significantly higher odds of exposure when compared with cattle ($p < 0.01$). We also observed adult animals having higher odds of exposure when compared to weaners and calves/kids/lambs. The estimated level of within-household clustering of *C. burnetii* exposures among livestock animals was 0.28 (95% CI: 0.18–0.37). The results of the LRT $\chi^2$ test used to assess the statistical significance of the two-way interactions terms of the independent variables in the final multivariable model showed that there were no statistically significant interaction effects. All the investigated two-way interaction product terms had p-values greater than 0.05.

The final multivariable mixed-effects model fitted for human data identified gender as the only important variable associated with *C. burnetii* seropositivity in humans (Table 5). Male individuals had significantly ($p = 0.03$) higher odds of testing positive for *C. burnetii* compared

**Table 4. Variables found to be associated with *C. burnetii* seropositivity in livestock using multivariable mixed-effects logistic regression model.**

| Variable | Category | Odds Ratio (95% CI) | SE | Z | P-value |
|---|---|---|---|---|---|
| Fixed effects | | | | | |
| Sex | Male | 1.00 (Ref.) | | | |
| | Female | 1.65 (1.13–2.42) | 0.19 | 2.60 | 0.01 |
| Age | Calf/kid/lamb | 1.00 (Ref.) | | | |
| | Weaner | 1.22 (0.34–4.40) | 0.66 | 0.30 | 0.761 |
| | Adult | 4.88 (1.46–16.29) | 0.62 | 2.57 | 0.01 |
| Species | Cattle | 1.00 (Ref.) | | | |
| | Goats | 3.99 (2.10–7.61) | 0.33 | 4.21 | < 0.01 |
| | Sheep | 4.02 (2.06–7.82) | 0.34 | 4.09 | < 0.01 |

Ref., reference category; CI, confidence intervals; SE, standard error.

Log likelihood = -908.50, number of observations = 2665, number of households = 152

The variance for the random effect variable (household ID) was 1.26 (95% CI: 0.87–1.45), SE = 0.68

**Table 5. Variables associated with *C. burnetii* seroprevalence in humans based on analysis using multivariable mixed-effects logistic regression model.**

| Variable | Category | Odds Ratio (95% CI) | SE | Z | P-value |
|---|---|---|---|---|---|
| Fixed effects | | | | | |
| Gender | Male | 1.00 (Ref.) | | | |
| | Female | 0.72 (0.53–0.98) | 0.15 | -2.12 | 0.03 |
| Family size | ≤5 | 1.00 (Ref.) | | | |
| | 6–10 | 0.88 (0.61–1.29) | 0.19 | -0.64 | 0.52 |
| | >10 | 0.66 (0.42–1.02) | 0.23 | -1.86 | 0.06 |

Ref., reference category; CI, confidence intervals; SE, standard error

Log likelihood = -535.90, number of observations = 970, number of households = 339

The variance for the random effect variable (household ID) was 0.093 (95% CI: 0.00–0.71), SE = 0.18

to females. Family size was forced in the final model as a fixed effect. Our comparison of the final model (comprised of gender and family size) and a model with gender only showed our final model had a lower AIC value (S1 Table), thereby showing the final model used in this study had a better fit. The estimated ICC for within-household correlation of *C. burnetii* exposures among humans was 0.03 (95% CI: 0.00–0.13). There was no significant interaction effect (LRT $\chi^2$ p = 0.53) observed for the two-way product term created for the covariates in the final model. The AIC values for all the multivariable models used to analyse livestock and human data, together with the variables included in each model are presented in S1 Table.

## Discussion

This study determined the serological exposure levels of *C. burnetii* among livestock and humans in Tana River county, and only humans in Garissa county. The detection of antibodies to *C. burnetii* among livestock and humans indicated that this pathogen is prevalent in the selected two counties which could significantly affect livestock and human health. The within-household (herd) correlation of *C. burnetii* exposure among livestock was high (ICC = 0.28). This could be due to frequent contact between infected and susceptible animals at the farm level or during grazing since this pathogen is extremely infectious. Infected animals also shed large quantities of *C. burnetii* in milk, faeces, urine, and vaginal discharges for months [13], which could enhance the transmission levels at the household through environmental contamination as the pathogen remains viable in the environment for a long time. For example, a previous study conducted in Washington and Montana (USA) detected large quantities of *C. burnetii* DNA in goats housing and birthing locations, and also small quantities in the air, a year following an outbreak [12].

In humans, we found a low ICC (0.03), an indication that human exposure to *C. burnetii* was not correlated among individuals within households. Furthermore, this study did not establish a positive association between herd-level seropositivity of *C. burnetii* and human exposure, consistent with a previous study done in Kenya [36]. Indeed, we found a large percentage of seropositive individuals (21.58%) who did not have a single animal in their herds that tested positive for *C. burnetii*. Whereas infected domestic animals are the primary sources for human infections [57], the lack of significant association between herd-level seropositivity of *C. burnetii* and human exposure in Tana River county suggests that seropositive individuals to this pathogen may have been exposed through different routes, besides the transmission from infected animals at the household level assessed in our analysis. In this study, the assessment of independent variables related to direct/indirect contact with animals and their products was also limited because data on other probable risk factors such as the consumption of

raw milk and its products; contact with reproductive fluids and tissues; the slaughter of animals, working in slaughterhouses, herding of livestock, sharing of houses with livestock, management of manure [8,21,40,41] and human tick bites [14,35,37], all known to be associated with the transmission of *C. burnetii* in humans, were not collected. It is also likely that these individuals may have been exposed through environmental contaminations or airborne dispersal that is a common occurrence in rural areas, mostly within the radius of less than 10 km away from infected herds, depending on stocking densities, wind speed/direction, and physical barriers such as vegetation [4]. However, this study could not determine the percentage of seroconversions among humans and livestock manifesting clinical disease or when these individuals/animals were infected since we tested animals and humans for the presence of antibodies which does not discriminate between past and active infections. Further incidence studies should therefore be conducted in the area to elucidate the main transmission routes of *C. burnetii* among humans.

Our study showed that the seroprevalance of *C. burnetii* in livestock differed significantly among animal age groups, with adult animals having higher seroprevalence estimates than weaners and calves/kids/lambs. This finding has been reported in other similar studies [37,58], and it is well known that older livestock could have had repeated exposure to the pathogen over their lifetime when compared to younger ones.

This study also found significantly higher seropositivity of *C. burnetii* among goats (15.22%) and sheep (14.22%) compared to cattle (3.0%), in agreement with previous studies done in Ethiopia [47], Nigeria [59] and Kenya [33]. However, other studies conducted in Egypt and Ivory Coast have reported higher seroprevalence estimates among cattle than goats and sheep [60,61]. The seroprevalence found in sheep was within the estimates of between 6.7% and 51.1% reported in a recent systematic review of Q fever in Kenya [27]. However, that of goats was relatively lower compared to the range of between 20 and 46% previously reported in the country [27,32]. Nonetheless, the estimated seroprevalence figures in sheep and goats were within the ranges (11–33%) reported among small ruminants across Africa [3]. For cattle, we found a relatively lower seroprevalence (3.0, 95% CI: 1.65–4.99) compared to other studies (range: 7.4–51.1%) conducted within Kenya [27,33,36]. Although this difference could be due to many factors including the use of different diagnostic kits, this finding agreed with most studies in Africa that have reported seroprevalence estimates of less than 13% [3]. The observed seroprevalence among humans (24.44%) was comparable to a study conducted in Egypt (25.71%) [62].

Based on gender, a significantly higher seroprevalence of *C. burnetii* was recorded in male individuals than females. The significant variation in seroprevalence by gender could be due to the different roles performed by men and women in the livestock value chain although this is culture-dependent. In general, men are involved in various livelihood activities such as the herding of livestock, working in abattoirs/slaughterhouses, assisting animals during birthing and slaughter which could increase their risk of exposure to this pathogen due to higher chances of encountering sick animals, contaminated viscera/fluids, aborted materials, or contaminated aerosols while grazing animals. A recent study on slaughterhouse workers in Western Kenya found that this occupational risk group had a higher seroprevalence of *C. burnetii* [34] compared with the general community in the same area [36]. Most of these workers were also male, highlighting the gender role in the transmission of *C. burnetii*. In addition, a recent systematic review also reported a high pooled seropositivity of *C. burnetii* among people working in abattoirs with individual studies recording seroprevalence estimates of 4.7–91.7% [8]. Comparably, another study done in Togo found higher seroprevalence estimates of *C. burnetii* among herders compared to non-herders [40]. In contrast, a significantly higher number of female adult animals had reactive antibodies against *C. burnetii* compared to adult males. This finding could be due to continued exposure of female adult animals to this pathogen over the

years as the market offtake of breeding females in pastoral areas tends to be lower compared to that of males. Nevertheless, caution is required in the interpretation of this finding since we sampled more female adult animals (n = 1708) than male adults (n = 291), which could have led to higher seroprevalence among female animals than males due uneven sex distribution.

Our study had several limitations including the testing of livestock/humans using ELISA rather than the complement fixation test (CFT) and immunofluorescence assay (IFA) which are considered as reference tests for diagnosis of *C. burnetii* infections in livestock and humans, respectively [5]. However, ELISA tests are more sensitive than both CFT and IFA [5,63]. The numbers of goats (1,333) and sheep (928) sampled were higher than those of cattle (466). This under-representation of cattle could have led to biased estimation of the overall seroprevalence of *C. burnetii* in livestock. Furthermore, serological cross-reactions have also been observed between *C. burnetii* and other bacteria such as *Bartonella* spp., *Legionella* spp., and *Chlamydia* spp. [64,65]. Therefore, there could have been an over-estimation of the seroprevalence estimates in humans and livestock due to false positivity. This study also collected data through a cross-sectional study design. Thus, we could not accurately determine current infections of *C. burnetii* in the targeted hosts since antibodies elicited against this pathogen persist for long after infection [66,67]. Therefore, we recommend future studies that have repeated sampling of individuals to obtain samples at both the acute and convalescent phases of disease progression.

## Conclusion

This study established that *C. burnetii* was prevalent in the study area as evidenced by the considerable seropositivity detected among livestock and humans. However, herd-level seropositivity of this pathogen was not significantly associated with human exposure. Among livestock species, both goats and sheep had significantly higher seroprevalence estimates compared to cattle. The detection of antibodies against *C. burnetii* in livestock and humans shows the need to establish One Health active surveillance in the area to identify the potential routes of transmission of this pathogen in humans. This finding also emphasizes the need to create more public awareness about this disease in the study area to reduce transmission and burden. This is also because previous studies in Kenya [68,69] and other African Countries [70,71] have shown that the adoption of food safety and biosecurity measures against zoonosis including Q fever, to be very low or non-existent among actors in the informal livestock value chain. In addition, this finding also suggests that Q fever should be considered as part of differential diagnosis when investigating non-malarial acute febrile illnesses in the study area. Further studies in the area are required to determine the socio-economic impacts of *C. burnetii* on livestock production and human health.

## Supporting information

**S1 STROBE Checklist.**
(DOC)

**S1 Data. Livestock data.**
(XLSX)

**S2 Data. Human data.**
(XLSX)

**S1 Table. AIC values for all the multivariable models used to analyse livestock and human data, together with the independent variables included in each model.**
(DOCX)

## Acknowledgments

We are grateful to all the participants in this study including farmers, veterinary officers and the local administrative officers from both Garissa and Tana River Counties. We also appreciate the clinical officers who were recruited from Ijara, Sangailu, Hola and Bura health centres for their contribution to human sampling. We thank Max Korir (ILRI) for creating the map used in this study.

## Author Contributions

**Conceptualization:** Damaris Mwololo, Daniel Nthiwa, Philip Kitala, Tequiero Abuom, Martin Wainaina, Salome Kairu-Wanyoike, Johanna F. Lindahl, Enoch Ontiri, Salome Bukachi, Ian Njeru, Joan Karanja, Rosemary Sang, Delia Grace, Bernard Bett.

**Data curation:** Damaris Mwololo, Daniel Nthiwa, Martin Wainaina, Salome Kairu-Wanyoike, Enoch Ontiri, Salome Bukachi.

**Formal analysis:** Daniel Nthiwa.

**Funding acquisition:** Delia Grace, Bernard Bett.

**Investigation:** Damaris Mwololo, Philip Kitala, Tequiero Abuom, Martin Wainaina, Salome Kairu-Wanyoike, Enoch Ontiri, Salome Bukachi, Ian Njeru, Joan Karanja, Rosemary Sang, Delia Grace, Bernard Bett.

**Methodology:** Philip Kitala, Tequiero Abuom, Martin Wainaina, Salome Kairu-Wanyoike, Johanna F. Lindahl, Delia Grace, Bernard Bett.

**Project administration:** Salome Kairu-Wanyoike, Johanna F. Lindahl, Delia Grace, Bernard Bett.

**Resources:** Delia Grace.

**Software:** Daniel Nthiwa.

**Supervision:** Philip Kitala, Tequiero Abuom, Salome Kairu-Wanyoike, Johanna F. Lindahl, Enoch Ontiri, Joan Karanja, Delia Grace, Bernard Bett.

**Validation:** Daniel Nthiwa.

**Visualization:** Daniel Nthiwa.

**Writing – original draft:** Daniel Nthiwa.

**Writing – review & editing:** Damaris Mwololo, Daniel Nthiwa, Philip Kitala, Tequiero Abuom, Martin Wainaina, Salome Kairu-Wanyoike, Johanna F. Lindahl, Enoch Ontiri, Salome Bukachi, Ian Njeru, Joan Karanja, Rosemary Sang, Delia Grace, Bernard Bett.

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
