## [Decision Letter · Decision Letter 0]

2 Aug 2021

Dear Nthiwa,

Thank you very much for submitting your manuscript "Serological epidemiological survey of Coxiella burnetii in livestock and humans in Tana River and Garissa Counties in Kenya." for consideration at PLOS Neglected Tropical Diseases. As with all papers reviewed by the journal, your manuscript was reviewed by members of the editorial board and by several independent reviewers. In light of the reviews (below this email), we would like to invite the resubmission of a significantly-revised version that takes into account the reviewers' comments. 

We cannot make any decision about publication until we have seen the revised manuscript and your response to the reviewers' comments. Your revised manuscript is also likely to be sent to reviewers for further evaluation.

Sincerely,

Claudia Munoz-Zanzi

Associate Editor

Fabiano Oliveira

Deputy Editor

Reviewer's Responses to Questions

**Key Review Criteria Required for Acceptance?**

**Methods**

-Are the objectives of the study clearly articulated with a clear testable hypothesis stated?

-Is the study design appropriate to address the stated objectives?

-Is the population clearly described and appropriate for the hypothesis being tested?

-Is the sample size sufficient to ensure adequate power to address the hypothesis being tested?

-Were correct statistical analysis used to support conclusions?

-Are there concerns about ethical or regulatory requirements being met?

Reviewer #1: -The study objectives are clearly defined and the methods in general clearly described and appropriate.

-line 120-121: Sample size calculation and sampling method is appropriate and clear, however authors state that study sites are "purposefully selected" but failed to explain why. References about the description of these counties do not clarify to the reviewers understanding, why these locations were selected for this study.

-line 234-235: Methods mention use of Multiple Correspondence Analysis (MCA), but without reference to the R package or method. Furthermore results of MCA are not discussed in results section. MCA can be useful as a variable reduction method, however, with the small number of variables in this study, multiple steps of variable reduction seems unnecessary, and most variables could have been investigated in multivariate model selection.

Reviewer #2: Line 134: Please justify the expected seroprevalences of C. burnetii in livestock and humans of 50% that were used. Same for the ICC of 0.3. Were these based on other studies? Also, please justify why you calculated the sample size for livestock in general rather than by species (i.e., separately for sheep, goats, and cattle). 

Line 170: How long were the samples stored? 

Laboratory testing: I suggest detailing more this section. Also, please report the inter- and intra-assay coefficients of variation for the ELISA tests, as well as the possible cross-reactivities with other pathogens. 

Lines 183-184: You mention that all samples were tested in duplicates. However, did you calculate a coefficient of variation between the duplicates, and was the sample repeated when there was a discrepancy? These details should be added. 

Lines 189-190: “We classified animals with PP of <30% as negative, suspect (borderline) if between 30% and 40% and positive if ≥40%”. How were these percentages chosen? Same question for the analysis of human samples.

Lines 203-204: “Samples that gave borderline results after re-testing were considered as negative during data analysis.” How many samples gave borderline results? Did you do the statistical analyses without these samples to check that you obtained the same results? 

Statistical analyses: Why did you not test any interactions in the models? Did you think about potential confounding factors? 

Line 217: How were the levels of this variable chosen?

Line 221: What is the difference between pastoralists and farmers? 

Lines 217 and 224-226: Why did you categorize these variables, which leads to a loss of information, rather than try transforming them (e.g., log-transformation, square-root, etc.)? Also please note that the assumption of normality is that the errors are normally distributed, not necessarily that all variables in the analysis are normally distributed. Please justify the choice of categories for age and family size. 

Line 235: You mention identifying non-correlated variables for multivariable analysis. Which variables were correlated? 

Line 240: What you are describing is the “backward” stepwise selection method not “forward-backward”. Please correct. 

Lines 243-244: Did you also check that residuals were normally distributed?

Reviewer #3: Study objectives are articulated with methods and the design is appropriate. Consider the following comments and suggestions: 

Line 134. Authors must detail why they assumed a seroprevalence of 50% in both populations (humans and animals) to calculate the sample size. 

Lines 139-141. Consider mentioning the rationale to select the Intra-cluster correlation coefficient (ICC) and the number of humans and animals in each household.

Lines 143-144. It is not clear the origin of the 169 and 129 selected households for human and animal sampling, respectively. Based on the experimental design, these are the selected clusters, thus is important to clarify the rationale behind these numbers.

Line 145. Is important to clarify the randomization method used (i.e., simple, systematic) 

Lines 151-152. Consider explaining, briefly, how did you manage with households without one or two of the target animal species. For example, households comprised only of cattle and goats but without sheep. Is important to avoid overrepresentation of one species above the others, which can bias the final results. 

Lines 173-179. It is not clear why authors did not include in the questionnaire variables related to C. burnetii transmission (e.g., raw milk consumption, contact with reproductive fluids and tissues, manure management, tick infestations, and others). Please, clarify this.

**Results**

-Does the analysis presented match the analysis plan?

-Are the results clearly and completely presented?

-Are the figures (Tables, Images) of sufficient quality for clarity?

Reviewer #1: -The results match the methods except for the lack of description of MCA results (even if briefly). Results are otherwise clear and comprehensive. Figures and tables match written results.

-line 254-257: Median number of animals per herd might be more useful instead of mean, with the mean given if very different from the median, which would indicate a skewed distribution of herd sizes. Median was actually given for human household size.

-line 259: When discussing topline numbers, it would be useful to have the response rate or know how many households refused to participate.

-tables 4 and 5: a general random effect coefficient is presented in the same column as the odds ratios. This is confusing as it is not interpreted in the same way, nor on the same scale. Individual household IDs would have a form of odds ratio, but there is no general random effect odds ratio.

Reviewer #2: Please add in the Supplementary material a table with the AIC values of all the models tested (both for livestock and humans), as well as the explanatory variables included. The explanatory variables included in the maximal and final models should be specified in the manuscript. 

Descriptive results: Please note that results are typically reported as means and standard deviations (SD) for normally distributed data and medians and interquartile ranges (IQR) or ranges for non-normally distributed data (see for example the paper by Habibzadeh F. “How to report the results of public health research”).

Line 268: remove “more”. 

Line 270: I suggest rephrasing to “The seroprevalence of C. burnetii differed significantly among animal age groups (χ2 = 73.45, df = 2, p < 0.01)”.

Table 1: In the column P>Z, is this the p-value? 

Table 2: In the columns P>χ2, are these the p-values?

Table 5: Why is family size in the final model if its effect is not significant? Also, family size is not mentioned in the corresponding paragraph of the results.

Reviewer #3: Consider the following comments and suggestions:

Line 257. It is not clear why 77 animals were sampled in some households; based on materials and methods (lines 151-152), the maximum number of animals in each selected household must be a maximum of 20.

Lines 252 and 260. Authors mention that animal and human samples were collected from 156 and 340 households, respectively. These numbers are significantly different from those mentioned in materials and methods (i.e., 169 and 129-lines 143-144-). Considering that households are the clusters used in the sampling design, authors should explain the reasons for these differences and if it could bias the interpretation.

Line 301. The term “Risk factor” is not accurate for a cross-sectional study, as you assess the exposure and the outcome at the same point in time. Is strongly recommended to use “associated factor” instead. Review this carefully throughout the text.

Figure 1. The figure must be improved to be self-explanatory for readers. Consider the following suggestions: 

• Include a panel with a map of Africa, showing Kenya's location. 

• In the small panel showing regions in Kenya put the name of Kenya and the neighboring countries.

• In the big panel showing the study areas put a North Arrow, make a grid with geographical coordinates in the margins, and put a scale bar at the bottom.

• In the big panel show the exact location of Bora, Hola, Ijara, and Sangailu. 

• Consider differentiating locations (e.g., different colors) where animal and human samples were obtained. 

Table 1. 

• Please, consider reducing the number of lines to separate columns and rows.

• Consider putting the 95% CI within brackets or parentheses next to the corresponding value (i.e., % Seroprevalence, Odds Ratio).

• Review the numbers in “Age” category. The total sum equals 2,665 and not 2,727 animals. 

Table 2.

• Please, consider reducing the number of lines to separate columns and rows.

• Consider putting the 95% CI within brackets or parentheses next to the corresponding value (i.e., % Seroprevalence).

• Review the numbers in “Age” category for sheep and goats. The total sums do not correspond to other categories.

• Please, clarify the numbers in the “herd/flock size” using a footnote. When you are reviewing the total number of animals in sheep and goats, within this category, you find numbers higher than 928 and 1333 animals, respectively. Is confusing. 

Table 3. 

• Please, consider reducing the number of lines to separate columns and rows.

• Consider putting the 95% CI within brackets or parentheses next to the corresponding value (i.e., % Seroprevalence).

• Review the numbers in the “Combined data from both Counties” category within the following subcategories: “Gender”, “Occupation”, “Own livestock”, “Land use”, and “Source of water”, they do not sum the 974 humans mentioned in the results. 

• Review the numbers in the “Tana River County” category within the following subcategories: “Gender”, “Occupation”, “Age”, “Own livestock”, “Source of water”, and “Herd exposure”, they do not sum the 484 humans mentioned in the results. 

• Review if the word “Clean” is appropriate in the “Herd exposure” variable. 

Table 4. 

• Please, consider reducing the number of lines to separate columns and rows.

• Consider putting the 95% CI within brackets or parentheses next to the corresponding value.

Table 5. 

• Please, consider reducing the number of lines to separate columns and rows.

• Consider putting the 95% CI within brackets or parentheses next to the corresponding value.

**Conclusions**

-Are the conclusions supported by the data presented?

-Are the limitations of analysis clearly described?

-Do the authors discuss how these data can be helpful to advance our understanding of the topic under study?

-Is public health relevance addressed?

Reviewer #1: Conclusions from the manuscript are mostly supported by the data presented as are limitations. Discussion however seems to be missing some comparisons to other seroprevalence for C. burnetii studies in Kenya. A quick Web of Science search with the words (burnetii) AND (kenya) led to 10 papers on this topic, none of which are referenced in this paper. Of note among these is a study with a very similar One Health design from 2016:

Wardrop NA, Thomas LF, Cook EAJ, deGlanville WA, Atkinson PM, Wamae CN, et al.(2016)The Sero-epidemiology of Coxiella burnetii in Humans and Cattle, Western Kenya: Evidence from a Cross-Sectional Study. PLoS Negl Trop Dis10(10):e0005032.doi:10.1371/journal.pntd.0005032

Another also sampled livestock and humans:

Knobel et al. 2013:Coxiella burnetii in Humans, Domestic Ruminants, and Ticks in Rural Western Kenya

Other studies focused on seroprevalence in ruminants:

Larson et al. 2019: The sero‐epidemiology of Coxiella burnetii (Q fever) across livestock species and herding contexts in Laikipia County, Kenya

Muema et al. 2016: Seroprevalence and Factors Associated with CoxiellaburnetiiInfection in Small Ruminants in Baringo County, Kenya

The other seroprevalence studies covered other species including wildlife, camels and ticks. A more comprehensive literature search might be necessary to better discuss the results presented in full context of the current knowledge in Kenya. This would also help in explaining how this study adds to this current knowledge already in existence.

Other comments include:

-line 368-370: specifying the two countries from ref 40, 41 would be better than "elsewhere", as is done in the next sentence with ref 42, 43.

-line 387-389: The authors might consider stratifying by age when looking at the effect of sex. As mentioned, adults are more likely to be positive compared to younger animals by virtue of having lived longer and so more time for exposure. Given the very uneven sex distribution in livestock, I would suspect that few males are adults compared to females, which would explain the higher sero-prevalence in females.

Reviewer #2: Discussion

Lines 375-377: “For cattle, we found a relatively lower seroprevalence compared to other studies (ranges; 7.4-51.1%) conducted within Kenya.” What are the possible reasons for this finding? 

Line 396: Please add a reference for the long-term persistence of C. burnetti antibodies.

Reviewer #3: Consider the following comments and suggestions:

Lines 346-347. When authors make the following statement: “Furthermore, this study did not establish a positive association between herd-level seropositivity of C. burnetii and human exposure.”, they must consider the limitations of a cross-sectional study to establish this kind of association. This could be also included in the discussion section. 

Lines 349-354. Authors affirm that: “Whereas infected domestic animals are the primary sources for human infections [37], the lack of significant association between herd-level seropositivity of C. burnetii and human exposure in Tana River County suggests that seropositive individuals to this pathogen may have been exposed through different routes, besides the direct contacts with infected animals and/or their products assessed in our analysis”. It must be emphasized that assessment of variables related with direct/indirect contact with animals and their products were limited. Questionnaires included categories like “Occupation” but did not inquire about other probable risk factors associated with transmission (e.g., consumption of raw milk or contact with animal feces). This should be considered in the discussion.

Lines 368-389. Authors should consider including in the discussion results from two recent studies conducted in the same region in Kenya. Firstly, the study made by Koka et al. (2018) [Koka et al. Coxiella burnetii Detected in Tick Samples from Pastoral Communities in Kenya. Biomed Res Int. 2018 Jul 9;2018:8158102. doi: 10.1155/2018/8158102. PMID: 30105251; PMCID: PMC6076967] which explores the circulation of C. burnetii in ticks collected from cattle, sheep, and goats. Secondly, the study published b Nyokabi et al. (2018) [Nyokabi et al. Informal value chain actors' knowledge and perceptions about zoonotic diseases and biosecurity in Kenya and the importance for food safety and public health. Trop Anim Health Prod. 2018 Mar;50(3):509-518. doi: 10.1007/s11250-017-1460-z. Epub 2017 Nov 13. PMID: 29130123; PMCID: PMC5818561.] which evidences the low levels of knowledge about zoonoses (including Q Fever) and low adoption of food safety and biosecurity measures.

Lines 390-393. Authors make a relevant approach to the study drawbacks, nonetheless, they should consider mentioning a drawback related to an absence of inquiry of variables associated with C. burnetii transmission, as mentioned in previous comments.

**Editorial and Data Presentation Modifications?**

Reviewer #1: no comments

Reviewer #2: (No Response)

Reviewer #3: Consider the following comments and suggestions:

Line 1. Review wording in the title. Consider using “Sero-epidemiological survey” instead of “Serological epidemiological survey”

Lines 14,15. “Department” instead of “Departrment”

Line 83. Consider writing “commonly manifest a febrile illness” instead of “commonly manifest with a febrile illness”

Lines 83-87. Consider including a brief description of the most common hospitalizations causes in those (or other) countries. Besides, it is recommendable to mention the neurological and cardiac compromise in some patients with acute infection. [See: Bernit, E., et al. (2002). Neurological involvement in acute Q fever. A report of 29 cases and review of the literature. Arch. Intern. Med. 162, 693– 700; Fournier, P.E., et al. (2001). Myocarditis, a rare but severe manifestation of Q fever report of 8 cases and review of the literature. Clin. Infect. Dis. 32, 1440–1447]

Lines 100-103. Probably, the list of pathogens related to febrile conditions in tropical Africa is larger. Authors are encouraged to review if other etiologies might be included. 

Line 104. Consider including at least one reference to support the relevance of the “One Health approach” to prevent and control zoonotic diseases like Q Fever. [e.g., Rahaman, M. R., et al. (2019). "Is a One Health Approach Utilized for Q Fever Control? A Comprehensive Literature Review." Int J Environ Res Public Health 16(5).]

Line 108. The text “and will inform the development of control strategies” is confusing. Review wording.

Line 113. “Institutional Animal Care and Use Committee” instead of “Institutional Animal Care and Uses Committee”

Line 115. Please, explain the abbreviation “AMREF”

Line 174. Consider reorganizing categories by putting first “species” and subsequently “age”. 

Lines 178-179. The text “and whether the sampled household owned livestock (yes or no) were also captured during sampling.” is confusing, consider rewording. 

Lines 189-190. Detail the source (or previous experience) for this classification system. 

Line 191. “Phase I” instead of “Phase 1”

Lines 199-200. Detail the source (or previous experience) for this classification system. 

Line 211. “seroprevalence” instead of “Seroprevalence”

Lines 215-216. Herd size is not a continuous variable, by its nature is a quantitative discrete variable.

Line 224. It is not clear why age and family size were recorded as continuous variables; they correspond to discrete variables. 

Lines 259-262. Consider including descriptive results related to occupation, and source of water as you mentioned these variables in materials and methods.

Lines 323-324. Review the term “predictor” in this phrase, as it is inaccurate. Consider using terms like “related” or “associated” based on the type of epidemiological study conducted. 

Line 343. Is suggested to put USA within parentheses. 

Line 364. Consider using “having higher seroprevalences than weaners…” or “having the highest seroprevalence.” instead of “having the highest seroprevalence than weaners…”.

Line 372. Please, add the citations of those previous studies with sheep in Kenya.

Line 376. “range” instead of “ranges”.

Lines 436-574. Review scientific nomenclature in all references.

**Summary and General Comments**

Reviewer #1: On the whole this is a well designed and straightforward study that answers its aims and objectives. The main weakness lies in failing to address both in the introduction and discussion, the current situation of data availability in Kenya for C. burnetii and how this study differentiates from previous sero-prevalence studies, improves the already existing knowledge base and guides future studies and intervention. Authors site the lack of epidemiologic knowledge (line 98) but do not mention the epidemiologic knowledge in existence. Similarly, in the conclusion (line 398), authors conclude that this study is evidence of prevalence in both humans and cattle, however this has already been shown in other papers and should likely not be the main takeaway from this study. As is, the novelty and thus to an extant the significance of this study are not clearly explained. 

This is a well designed and executed epidemiology study that would clearly improve knowledge of this disease in Kenya, but should be better presented in context of the already existing knowledge.

Reviewer #2: This study presents the results from a serological survey of Coxiella burnetti exposure in livestock and humans in two counties in Kenya. This is an interesting study, which provides important data on a pathogen that has major impacts on human and animal health. However, I have some concerns regarding the sample size calculations, as well as the laboratory and statistical analyses. Please see my detailed comments and suggestions. 

Abstract

Line 37: I suggest adding the word “respectively” after humans as household ID is the random effect in both the livestock and the human models. 

Lines 45-46: “The results from multivariable model showed that sex (female), age (adults), and species (goats and sheep) were significant factors associated with exposure to C. burnetii in livestock.” This sentence is unclear. I think that you meant that higher seroprevalences were found in females, adults and sheep and goats, I suggest reformulating. 

Line 48: Replace “with” with “than”. 

Introduction

The first two paragraphs of the introduction (lines 66-95) could be shortened. 

Line 101: I suggest replacing “febrile-causing” with “fever-causing” or reformulating. 

Lines 104: I suggest detailing in the introduction what a “One Health approach” is and entails, and why such approaches are important. You could even give examples (see the book “One Health Case Studies: Addressing Complex Problems in a Changing World”).

Reviewer #3: The submitted study corresponds to an observational cross-sectional survey of Coxiella burnetii in livestock and humans from two regions in Kenya. The manuscript is well written and presents comprehensively the background, methods, and results. All relevant data is provided and, in general, STROBE guidelines were followed. The obtained results are relevant for this region and complement diverse studies in this neglected disease in Kenya. Is unfortunate the absence of animal data in Garissa County to be related to human variables and results. Despite authors mentioned some drawbacks in the text, they should consider reviewing the lack of exploration and analyses of variables related to animal and human transmission. Do not discard to include a brief text to discuss probable biases in the study.

PLOS authors have the option to publish the peer review history of their article (what does this mean?). If published, this will include your full peer review and any attached files.

Reviewer #1: No

Reviewer #2: No

Reviewer #3: No
---

## [Decision Letter · Decision Letter 1]

20 Dec 2021

Dear Nthiwa,

Thank you very much for submitting your manuscript "Sero-epidemiological survey of Coxiellaburnetii in livestock and humans in Tana River and Garissa Counties in Kenya." for consideration at PLOS Neglected Tropical Diseases. As with all papers reviewed by the journal, your manuscript was reviewed by members of the editorial board and by several independent reviewers. The reviewers appreciated the attention to an important topic. Based on the reviews, we are likely to accept this manuscript for publication, providing that you modify the manuscript according to the review recommendations. 

Thanks for addressing the reviewer's comments. Reviewers added a few more helpful comments that will improve the clarity of the final version.

Sincerely,

Claudia Munoz-Zanzi

Associate Editor

Fabiano Oliveira

Deputy Editor

Thanks for addressing the reviewer's comments. Reviewers added a few more helpful comments that will improve the clarity of the final version.

Reviewer's Responses to Questions

**Key Review Criteria Required for Acceptance?**

**Methods**

-Are the objectives of the study clearly articulated with a clear testable hypothesis stated?

-Is the study design appropriate to address the stated objectives?

-Is the population clearly described and appropriate for the hypothesis being tested?

-Is the sample size sufficient to ensure adequate power to address the hypothesis being tested?

-Were correct statistical analysis used to support conclusions?

-Are there concerns about ethical or regulatory requirements being met?

Reviewer #1: (No Response)

Reviewer #2: No comments.

Reviewer #3: The revised version included most of comments and suggestions made for the original submission. Authors provided a detailed explanation through their responses to each reviewed item.

**Results**

-Does the analysis presented match the analysis plan?

-Are the results clearly and completely presented?

-Are the figures (Tables, Images) of sufficient quality for clarity?

Reviewer #1: (No Response)

Reviewer #2: No comments.

Reviewer #3: The revised version included most of comments and suggestions made for the original submission. Authors provided a detailed explanation through their responses to each reviewed item.

**Conclusions**

-Are the conclusions supported by the data presented?

-Are the limitations of analysis clearly described?

-Do the authors discuss how these data can be helpful to advance our understanding of the topic under study?

-Is public health relevance addressed?

Reviewer #1: (No Response)

Reviewer #2: No comments.

Reviewer #3: The revised version included most of comments and suggestions made for the original submission. Authors provided a detailed explanation through their responses to each reviewed item.

**Editorial and Data Presentation Modifications?**

Reviewer #1: (No Response)

Reviewer #2: No comments.

Reviewer #3: Line 63. Put a period after “Kenya”. 

Line 83. Consider omitting “of the bacteria” after “tick”

Lines 126-128. Authors might consider relocating the following text: “Our study showed that C. burnetii was circulating among livestock and human populations and provides a basis for the establishment of integrated livestock-human surveillance in the area” within the discussion or conclusion sections, where it fits better

Line 188.Is confusing the mean of “unreliable security”, is that related to “safety concerns”? Consider rewording to make it clearer.

Lines 356-358. Consider including the p-value for male and female seropositivity comparison. 

Lines 453-456. Review wording for this phrase, is confusing.

**Summary and General Comments**

Reviewer #1: The reviewer is satisfied with the responses to the previous comments.

Reviewer #2: Thank you for your detailed responses and clarifications. All your revisions were very thorough. I have just a couple minor suggested edits (line numbers correspond to the version of the manuscript with track changes): 

- Line136: I suggest replacing "planned to assess" with "assessed". 

- Lines 155-157: "The above study sites were purposefully selected to allow the estimation of the risk of transmission of Coxiella burnetii between livestock and humans." It is still unclear why these specific study sites were chosen rather than others. What are the characteristics that rendered them particularly relevant to study? 

- Line 204: I suggest replacing "all the three-targeted species" with "all three targeted species". 

- Line 228: I suggest specifying after "in 2016" the number of months/years that had passed since the samples were collected. 

- Line 406: Replace "for" with "as". 

- Lines 495-499: I suggest rephrasing the sentence "Further incidence studies that use multiple serological tests to improve net diagnostic performance as well as changes in antibody titres, as well as molecular tests to detect active infections should therefore be conducted in the area to elucidate the main transmission routes of C. burnetii among humans." It is unclear and difficult to follow. 

- Line 519: I suggest replacing the word "study" with "difference".

Reviewer #3: I must acknowledge authors for the comprehensive review and considering most of suggestions and comments made in the original submission. Also for providing a detailed explanation through their responses to each reviewed item. The submitted revision enhanced its global quality and clearness. Also the new map and tables are self-explanatory, and they improved considerably. Some minor suggestions are listed to be considered for a final version.

PLOS authors have the option to publish the peer review history of their article (what does this mean?). If published, this will include your full peer review and any attached files.

Reviewer #1: No

Reviewer #2: No

Reviewer #3: No

Figure Files:

Data Requirements:

Reproducibility:

References

---

## [Editor Report · Decision Letter 2]

28 Jan 2022

Dear Nthiwa,

We are pleased to inform you that your manuscript 'Sero-epidemiological survey of Coxiella burnetii in livestock and humans in Tana River and Garissa Counties in Kenya.' has been provisionally accepted for publication in PLOS Neglected Tropical Diseases.

Best regards,

Claudia Munoz-Zanzi

Associate Editor

Fabiano Oliveira

Deputy Editor

---

## [Editor Report · Acceptance letter]

1 Mar 2022

Dear Nthiwa,

We are delighted to inform you that your manuscript, "Sero-epidemiological survey of *Coxiella burnetii* in livestock and humans in Tana River and Garissa Counties in Kenya. ," has been formally accepted for publication in PLOS Neglected Tropical Diseases.

Best regards,

Shaden Kamhawi

co-Editor-in-Chief

Paul Brindley

co-Editor-in-Chief
